# Hyaluronan with Different Molecular Weights Can Affect the Gut Microbiota and Pathogenetic Progression of Post-Intensive Care Syndrome Mice in Different Ways

**DOI:** 10.3390/ijms24119757

**Published:** 2023-06-05

**Authors:** Lu Li, Yuanyuan Jiang, Qianqian Zhu, Dawei Liu, Mingkai Chang, Yongzhe Wang, Ruitong Xi, Wenfei Wang

**Affiliations:** Biopharmaceutical Lab., College of Life Science, Northeast Agricultural University, Harbin 150030, China; lilu021302@163.com (L.L.); jyy580@126.com (Y.J.); z13356203394@163.com (Q.Z.); ldw1006@outlook.com (D.L.); 18537986157@163.com (M.C.); 13805409508@163.com (Y.W.); tong020401@126.com (R.X.)

**Keywords:** PICS, hyaluronan, gut microbiota

## Abstract

Post-intensive care syndrome (PICS) poses a serious threat to the health of intensive care unit (ICU) survivors, and effective treatment options are currently lacking. With increasing survival rates of ICU patients worldwide, there is a rising interest in developing methods to alleviate PICS symptoms. This study aimed to explore the potential of using Hyaluronan (HA) with different molecular weights as potential drugs for treating PICS in mice. Cecal ligation and puncture (CLP) were used to establish a PICS mice model, and high molecular weight HA (HMW-HA) or oligo-HA were used as therapeutic agents. Pathological and physiological changes of PICS mice in each group were monitored. 16S rRNA sequencing was performed to dissect gut microbiota discrepancies. The results showed that both molecular weights of HA could increase the survival rate of PICS mice at the experimental endpoint. Specifically, 1600 kDa-HA can alleviate PICS in a short time. In contrast, 3 kDa-HA treatment decreased PICS model survivability in the early stages of the experiment. Further, via 16S rRNA sequence analysis, we observed the changes in the gut microbiota in PICS mice, thereby impairing intestinal structure and increasing inflammation. Additionally, both types of HA can reverse this change. Moreover, compared to 1600 kDa-HA, 3 kDa-HA can significantly elevate the proportion of probiotics and reduce the abundance of pathogenic bacteria (*Desulfovibrionaceae* and *Enterobacteriaceae)*. In conclusion, HA holds the advantage of being a potential therapeutic drug for PICS, but different molecular weights can lead to varying effects. Moreover, 1600 kDa-HA showed promise as a protective agent in PICS mice, and caution should be taken to its timing when considering using 3 kDa-HA.

## 1. Introduction

Post-intensive care syndrome (PICS) is a complex condition that poses a serious health risk to critically ill patients in recovery. According to an official statement by the American Thoracic Society, more than half (50%) of those who survive a stay in the intensive care unit (ICU) will experience at least one of the symptoms related to PICS [1]. Furthermore, one-third of PICS patients are at risk of returning to the ICU or even dying within the first year of recovery [2,3]. In recent years, with the viral pandemics such as COVID-19, millions of people around the world receive ICU treatment each year. Although most people can survive through treatment, the subsequent development of PICS remains a significant impediment to patients’ long-term health [4]. Therefore, relief of PICS symptoms is essential to help ICU survivors recover as well as improve their quality of life.

The intestinal system is the largest endocrine organ and the second largest immune system in the human body, closely linked to nutrient intake, immune regulation, and metabolism of substances and energy. Critically ill patients often experience dysfunction of the intestinal system due to ischemia resulting from sepsis, severe COVID-19, or other major illnesses [5]. This leads to a weakened intestinal barrier function, which in turn facilitates the translocation of gut bacteria or toxins and ultimately leads to endogenous damage to multiple organs [6]. At the same time, the ecosystem of the gut microbiota is also disrupted during this period, resulting in the dysregulation of the microbiota structure and function, further impairing intestinal function and exacerbating the patient’s condition [7]. Jinlin Zhang et al. proposed that damage to the structure and function of the intestinal system, as well as the disruption of the gut microbiota ecosystem, are important factors in the cycle of health deterioration in PICS patients (including intestinal function-immune regulation and catabolism) [5]. Based on this viewpoint, we hypothesis that the interventions targeting the gut microbiota to maintain intestinal system stability may be an effective means of improving PICS.

Hyaluronic acid (HA) is a biologically versatile substance found widely in animals which can regulate immune function [8] and improve the intestinal system [9]. These functions are in line with comprehensive needs of PICS patients. Further, as a structural element in animals, HA has relatively mild effects, and its oral safety has been verified [8,10]. Therefore, we speculate that HA is a potential agent for the treatment of PICS. However, it should be noted that, although it has the same structural units, HA with different molecular weights often show very different [11,12] or even opposite biological activities [13]. Therefore, whether HA can relieve PICS symptoms and whether HA of different molecular weights have the different effects on PICS treatment still need to be explored.

To investigate the effect of different molecular weight HA on PICS, we followed the method set up by Amanda M. Pugh to establish a PICS mouse model using cecal ligation and puncture (CLP) [14]. We intervened and treated the PICS model mice with 1600 kDa HMW-HA and 3 kDa oligo-HA, respectively, via gavage. We analyzed the effects of 1600 kDa-HA and 3 kDa-HA on PICS mice by observing their survival rate and physiological condition. Further, we also investigated the role of the gut microbiota in the relief of PICS symptoms with HA of different molecular weight. Our findings may provide evidence for a novel therapeutic strategy for the prevention of PICS.

## 2. Results

### 2.1. The Effects of Oral Administration of 3 kDa-HA and 1600 kDa-HA on the Survival Rate, Body Weight, and Grip Strength of PICS Mice

According to the survival curve, the survival rate of both the 1600 kDa-HA- and 3 kDa-HA-treated mice was higher than that in the CK group on the 30th day after treatment. This result demonstrated that HA can alleviate CLP-induced PICS symptoms in mice. Interestingly, different molecular weights were significantly different in the intervention effect of HA on PICS, and the survival rate of the 3 kDa-HA-treated mice was even lower in the early stage of the experiment than that in the CK group.

Among the examinations recommended in evaluations of PICS, grip strength is one of the most useful physical examinations, as it is easily assessed, and it can be performed to detect slight muscle weakness as a continuous variable of muscle strength. In addition, Nakamura et al. also proved that grip strength is a useful evaluation that also reflects the mental status in PICS [15]. Therefore, grip strength was chosen in this study to evaluate the intervention effect of HA on PICS mice model. The grip strength test results on the 30th day of administration also showed significant differences between the 1600 K and 3 K groups, while the 1600 K group was closer to the WT group (Figure 1C). During the experiment, the weight gain of the mice in each group was relatively stable, with no significant differences.

### 2.2. Effect of Oral Administration of 3 kDa-HA and 1600 kDa-HA on Lung Pathological Changes and Fibrosis Degree in PICS Mice

The pulmonary pathological changes of PICS mice were observed at the 10th (Figure 1D) and 30th (Figure 1E) day. We defined the results on the 10th day as short-term treatment results and the results on the 30th day as long-term treatment results. According to the H&E staining results, it can be clearly seen that the lung lesion severity gradually deepened, and the lung tissue structure was disrupted in the CK group with the prolongation of the disease time. The alveolar wall was significantly thickened, with a large number of inflammatory cells infiltrating the alveoli and interstitial spaces, accompanied by alveolar hemorrhage, alveolar wall damage, and other obvious pathological features. The lungs of PICS mice in the 3 K group showed alveolar hemorrhage and alveolar wall thickening on the 10th day, but the lesions improved by the 30th day. In contrast, no significant lesions were observed in the 1600 K group. The Masson results (Figure 1F,G) also showed that the degree of pulmonary fibrosis in each group did not change significantly at each time point.

### 2.3. Morphology and Pathological Changes of Intestinal Tissues in Each Group of Mice

After 10 days of administration, three mice from each group were randomly selected for intestinal morphology observation (Figure 2A). The CK group showed obvious atrophy in the intestinal structure, which was more prone to damage, and the intestinal wall was swollen and dark, which may be due to long-term insufficient perfusion or intestinal obstruction. Although there was also swelling in the 3 K and 1600 K groups, the overall structure was smooth, intact, and tended to be normal.

The proximal and distal colon of the mice were removed for H&E and Masson staining. On the 10th day, the intrinsic muscle layer of the CK group shrunk, the thickness decreased significantly, the goblet and absorptive cells in the crypt decreased, and the top of the crypt structure was clearly damaged, while the 3 K group showed semi-lunar fold loss. However, after 30 days of administration, the intestinal structure of the 3 K group significantly recovered, while that of the CK group continued to deteriorate. The 1600 K group showed good recovery status, significantly better than other groups on the 10th day, and the intestinal structure in later stages tended to be similar to that of the WT group.

### 2.4. Number of Mice Gut Microbiota and Community Diversity

After the sequencing and clustering analysis of the obtained sequences, it was found that the number of microbial OTUs in the 1600 K group gradually recovered to normal levels, but the 3 K group had significantly fewer OTUs compared to other PICS groups (Figure 3A,B). However, its alpha diversity results were closer to the WT group compared to the 1600 K and CK groups (Figure 3C,D).

Beta diversity analysis was performed on the community composition and phylogenetic information of multiple samples. As shown in Figure 3E, before administration, the gut microbiota of the PICS groups and the normal mice were significantly different. However, the community structure of the 1600 K and CK groups tended to recover towards the WT group by the 30th day. However, the 3 K group showed different results in different classifications and algorithms. As the complexity of the sample composition increased, the 3 K group showed a tendency towards the WT group in specific classification standards but deviated from the WT group in other classification standards. This requires further analysis of the microbial community structure of each group.

### 2.5. Structure and Abundance of Mouse Gut Microbiota

In this experiment, about five bacterial phyla were detected in the mice intestinal. The gut microbiota structure of healthy mice was relatively stable, with Bacteroidetes and Firmicutes being the dominant phyla. However, in the gut of PICS mice, Bacteroidetes lost its dominant position and was replaced by Proteobacteria, whose proportion increased significantly. This phylum tends to accumulate when the gut immune system is abnormal [16]. The abundance of the microbial community at the phylum level indicates that the gut microbiota structure of PICS mice is severely disturbed. As the mice in each group recovered, the dominant position of Bacteroidetes gradually recovered, and Proteobacteria decreased. Among them, the Proteobacteria ratio in the 3 K group decreased the quickest, but on the 10th day, the 1600 K group showed a greater proportion of Verrucomicrobia—a probiotic phylum that can regulate metabolic disorders in the body.

As shown in Figure 4A,B, Proteobacteria, Bacteroidetes, and Firmicutes act an crucial role on the composition and function of gut microbiota in each group. Therefore, we also focused on the abundant changes in microbial taxa at different taxonomic levels contained in these phyla in each group, and the evolutionary relationships between microbes are shown in Figure 4A,B.

Although Firmicutes has always been dominant, the species composition of the Firmicutes changes significantly in different groups. The proportion of Erysipelotrichia in the 3 K and 1600 K groups was low, while in the CK group, the proportion of this bacterial genus was even significantly higher than in the WT group at the end of the experiment. After 30 days of treatment, the proportion of Clostridium_sensu_stricto in the 3 K group was significantly higher than that in the CK and 1600 K groups. The proportion of Blautia was lower in the 1600 K and 3 K groups than in the CK group, and the proportion was even lower in the 1600 K group. The significant differences in Bacilli were Lactobacillus and Lactococcus. At the beginning of the experiment, the proportion of Lactobacillus in all groups was significantly lower than in the WT group, but the growth rate of proportion in the 3 K group was the slowest, while that in the CK group was the quickest in the later period. Although the proportion of Lactococcus in PICS groups was lower than WT groups, the 3 K group was obviously higher than the other two groups. However, at the genus level, the proportion recovery of the 1600 K group was better than 3 K group.

Before administration, the proportion of Bacteroidetes in all PICS model groups was significantly lower than that in the WT group, and their proportion increased in the later stage of the experiment. Among them, the recovery rate of Prevotella in the 3 K group was significantly higher than in the other two groups, while the recovery rate in the CK group was the slowest. After medication, the proportion of Bacteroides was significantly higher in the 1600 K group than in the CK group on the 10th day, and the proportion in the 3 K group was higher than in the CK group on the 30th day.

The changes in Deltaproteobacteria in Proteobacteria are most obvious in the 3 K group, with varying degrees of decrease compared to the CK and WT groups. Helicobacter in Epsilonproteobacteria had a lower proportion in the CK group than in the WT group and a higher proportion in the 1600 K and 3 K groups. Enterobacteriales in Gammaproteobacteria underwent the most complex changes. Serratia experienced the fastest decrease in proportion in the 1600 K group, while Escherichia_Shigella showed almost no change in the CK group during the experimental period. The proportions in the 3 K and 1600 K groups decreased, with the 3 K group eventually approaching the WT group. The CK group had more unclassified_Erysipelotrichaceae than the WT group.

In addition, Actinobacteria is a phylum that increases in the 1600 K group after medication, but it is almost not reflected in the 3 K and CK groups. The proportion of Candidatus_Saccharibacteria in the 3 K group is lower than that in the CK group.

### 2.6. Mice Intestinal Microbiota Functional Analysis

Using PICRUSt to interpret the potential functions of the microbiota composition data then clustered and analyzed the data. The KO (Figure 5A) and COG (Figure 5B) results show that the functional differences between the groups mainly exist in material energy metabolism, maintenance of biological structures, drug transport, antimicrobial peptide transport, and other aspects. However, the functional profile of the microbiota in the 3 K group is significantly different from the other groups (Figure 5C,D). The analysis of COG results using String (Figure 5E,F) revealed that the enriched pathways are most associated with antimicrobial peptide transport systems, and plenty of pathways relating to metabolism, synthesis, and transport are closely related to therapeutic effect.

## 3. Discussion

Despite sufficient attention being given to post-intensive care syndrome (PICS) in recent years, research on its pathogenesis remains insufficient, and clinical treatment options are lacking. Although researchers have attempted to treat PICS using antibiotics [17,18] or anti-catabolic therapy [5], these approaches have not fully resolved the complex symptoms of immune system disorders, catabolic disorders, and insufficient nutritional intake [19]. The administration of stimulant drugs may even induce endogenous damage in patients [18]. So, it is necessary to tailor the most appropriate therapeutic strategy for patients with PICS.

Hyaluronic acid (HA) is a ubiquitous structural substance in animals. Its molecular weight greatly influences its biological activity. HMW-HA refers to a group of HAs with the largest molecular weight, while oligo-HA refers to a group of HAs with the smallest molecular weight. These two groups have completely different effects on immune regulation [12]. In this study, we used 1600 kDa-HA and 3 kDa-HA, representing HMW-HA and oligo-HA, respectively, to intervene in PICS mice. The feasibility of using HA to alleviate PICS disease was preliminarily verified, and the difference in intervention effects of the two HAs was compared.

Clinical data suggest that patients with PICS have the highest mortality rate in the early stages, namely, within the first year, which gradually decreases later [20]. The persistent state of immune dysregulation in patients renders them more vulnerable to exogenous infections or endogenous damage [19]. In this study, it was observed that untreated PICS mice had a high risk of death even after the acute phase (Figure 1A), concentrated within 10 days. Surviving mice, even if not dead, had severely compromised health, which was difficult to recover to normal levels for a long time and had severe lung lesions (Figure 1D–G). After oral gavage with 3 kDa-HA or 1600 kDa-HA, the condition of PICS mice was improved. However, the onset time of the two was different. Treatment with 1600 kDa-HA reduced the mortality rate of mice and significantly improved the lung lesions of mice in the short time. The effect of 3 kDa-HA had no significant variation with CK group over a short time, but there was a reduction in lung lesions over a long time. These results indicate that oral administration of 1600 kDa-HA is more advantageous in rapidly improving the condition in PICS mice. However, the differential results caused by the length of the 3 kDa-HA intervention also attracted our attention.

The deterioration of intestinal function is the key factor that drives the worsening of patients’ conditions [5]. There is a vicious cycle in patients with PICS, in which the continuous deterioration of the intestinal system seriously affects their health [5]. The deterioration of the intestinal system in the PICS mice was severe (Figure 2). However, under the intervention of 3 kDa-HA or 1600 kDa-HA, the changes in the intestinal system were particularly noteworthy. The colonic recovery of mice in the 1600 K group was extremely apparent over a short time (Figure 2B,D). However, in the 3 K group, the colonic region was significantly aggravated via a short period of administration. Nevertheless, the results after 30 days of administration showed that the colonic region in the 3 K group had improved (Figure 2C,E). As the changes in the gut microbiota conform to a chronic process of change, the improvement in the 3 K group may be attributed to the interaction between 3 kDa-HA and the gut microbiota, which has been gradually improving via a long period of intervention.

The microbial community inhabiting the hosts gut is closely related to the stability of gut structure and function [21,22]. Bacteroidetes and Firmicutes are the dominant phyla in a healthy human body. Proteobacteria, Actinobacteria, and Candidatus_Saccharibacteria also play important roles in the structure and function of the gut microbiota [23]. Our study revealed that the gut microbiota structure and abundance of PICS mice were abnormal (Figure 4C,D). Prior to HA intervention, Proteobacteria replaced Bacteroidetes as the second most abundant phylum in the gut, after Firmicutes (Figure 4A). This is often observed in patients with acute enteritis and other acute illnesses and indicates gut environment deterioration and abnormal function [24]. In the duration of therapy, the gut microbiota structure of all groups of PICS mice showed recovery due to the organism’s powerful self-recovery ability. However, there was significant differences in the degree of recovery between the groups. In the CK group, the proportion of Proteobacteria remained significantly high until the end of the experiment, while the proportions of Proteobacteria in the two groups treated with HA were significantly closer to those in the WT group.

This finding draws attention to the phylum Proteobacteria, which contains many pathogenic bacteria. *Desulfovibrionaceae* and *Enterobacteriaceae* are typical pathogenic bacteria in this phylum. Pitcher, M.C. demonstrates that *Desulfovibrionaceae* can increase sulfide production and generate hydrogen sulfide, which can cause mucosal damage in patients with ulcerative colitis [16,25]. *Enterobacteriaceae* are often invasive bacteria that accumulate in large quantities during intestinal diseases [26]. *Klebsiella* in *Enterobacteriaceae* can cause FNF release, thereby stimulating the immune system [27,28]. Multiple studies, including those by Martin, H.M., and Sepehri, S., have suggested that *Escherichia* promotes intestinal inflammation and cancer development and that this genus can stimulate macrophages to recognition them and induce gut immune responses [29,30]. All of these factors can lead to a deterioration of gut function and structure and even further immune imbalance in patients. *Serratia* can even interact with the gut immune system. When its proportion increases, it can reduce the host’s immune response. A. Samuelsson also pointed out that this genus is resistant to drugs and can be transmitted between patients, causing bacterial infections [31,32]. These findings also tallied with our experiment. The CK group of mice showed a significantly higher proportion of these strains and slower recovery rates compared to the treatment group. The *Desulfovibrionaceae* and *Enterobacteriaceae* proportions were significantly higher in the CK group compared to the WT group. The proportions of these two phyla in the 1600 K and 3 K groups gradually recovered, with the 3 K group being closer to the WT level. These findings suggest that the gut system of CK mice is in a prolonged state of immune imbalance, and it is difficult for it to recover to a healthy state on its own. However, HA intervention can reduce the abundance of pathogenic bacteria, especially in the 3 K group.

In addition, the abnormal enrichment of *Blautia*, a member of the Firmicutes phylum, in the CK group can lead to intestinal mucosal damage and disrupt the intestinal barrier [33]. These alterations in the microbiota are one of the reasons why the control group’s intestinal structure is damaged and has not been able to repair itself. These changes in the gut microbiota also contribute to the deterioration of the gut function–immune system-metabolic cycle.

Comparing the results of each group, we found that the proportion of probiotics in the 3 K group was significantly higher than that in the CK group, indicating that 3 K intervention had a positive effect on promoting the growth of beneficial bacteria in the gut. Although the 1600 K group was also improved compared with the CK group, the proportion of beneficial bacteria was still lower than that in the 3 K group. *Clostridium* can produce multiple beneficial acids [34]. *Prevotella* can produce short-chain fatty acids, lower cholesterol levels, and enhance the metabolic rate of people with high energy intake [35,36]. Eric M. Brown et al. found that *Bacteroides* can produce sphingolipids, thereby regulating the inflammatory response of patients [37]. These are all beneficial bacteria that are essential for maintaining the stability of the intestinal system. *Lactococcus* is a well-known probiotic that is widely used in food fermentation and even in the medical field. This probiotic can participate in the regulation of the body’s immune system through various pathways and regulate the structure of the gut microbiota [38,39]. Although there was no significant improvement in these microbiota changes during the short-term 3 K administration, long-term intervention resulted in a significant increase in the proportion of probiotics in the gut of the 3 K mice and a significant decrease in the proportion of harmful bacteria. As a result, the proportion of the gut microbiota in the 3 K group became more similar to that of the WT group, which is evident from the alpha diversity of the microbiota (Figure 3C–E). This confirms our hypothesis that 3 K has the potential to regulate the structure and function of the gut microbiota in PICS mice.

However, comparing the major enriched microbiota among the groups, although the ratio of beneficial and harmful bacteria in the 3 K group is more coordinated, and the overall ratio of microbiota tends to be closer to the WT group, the diversity of microbiota is significantly lower than that of other groups (Figure 3A,B). Previous studies have shown that only a fraction of gut microbiota, ranging from 10% to 25%, can be studied in depth, and many bacteria that have a small proportion but a significant impact on maintaining the structure and function of microbiota may be difficult to detect [40]. The absence of these bacteria may indirectly affect human health. The proportion of probiotics in the 1600 K group was lower than that in the 3 K group, but the structure and function of microbiota in the 1600 K group may be more stable than those in the 3 K group, and chronic regulation may promote the body’s gradual recovery. Furthermore, as a protective group, the abundance of Firmicutes in the 1600 K group was significantly higher than that in other groups at the end of the experiment. Even at the short time of the experiment, Verrucomicrobia and Actinobacteria, two microbiota phyla closely related to metabolism regulation [41,42], were detected in the 1600 K group, which were almost undetectable in the other two PICS groups. The increase in their abundance may also be a contributing factor to the slightly lower proportion of probiotics in the 1600 K group compared to the 3 K group. However, the reduction in the abundance of 3 K group bacteria needs to be further explored

Abnormal muscle breakdown is a significant manifestation of metabolic syndrome, which is one of the main symptoms of PICS [2,5]. Upon conducting the predictive analysis of their functions, we discovered that the metabolic pathways of the CK and 3 K groups were more active than the other groups. Although 1600 K group showed less activity in metabolic pathways, this group showed significant recovery of absorption cells in the intestinal tract, and the grip strength of mice on the 30th day was also close to the WT group (Figure 1C). In contrast, the 3 K group showed completely opposite results, with an increase in metabolic level, but the recovery of absorption cells was not as good as in the 1600 K group. Hence, we speculate that the substances were consumed more than they were ingested in the 3 K group. This may explain why the grip strength recovery of the 3 K group was inferior to that of the 1600 K group, despite having a better gut microbiota structure.

As an HMW-HA, 1600 kDa-HA has been shown to demonstrate anti-inflammatory, wound healing, and gut system reconstruction effects [43,44,45]. In this study, the rapid recovery observed in the 1600 K group of mice may be particularly attributed to the direct effects of 1600 kDa-HA through repairing intestinal damage, regulating inflammatory responses, and then subsequently affect changes in gut microbiota and patients’ overall health. Through this direct repair function, 1600 kDa-HA can quickly and effectively alleviate PICS symptoms in mice. In contrast, 3 kDa-HA, as an oligo-HA, has opposite biological activity to HMW-HA [13]. However, in this study, the short-term and long-term effects of 3 kDa-HA on PICS mice showed significant differences, and 3 kDa-HA also demonstrated a significant improvement in gut microbiota structure. This leads us to speculate that 3 kDa-HA may not have a direct effect on improving PICS symptoms in patients, but it may indirectly affect patients’ health by regulating gut microbiota. However, the specific effects of these two types of HA in improving PICS symptoms require further investigation.

## 4. Materials and Methods

### 4.1. Animal Experiment

Healthy female KM mice (aged six weeks) were purchased from Changchun Yisi Experimental Animal Technology Co., Ltd. (Changchun, China). All mice were housed under controlled environment (room temperature of 23 ± 2 °C with 12/12 h light/dark cycles) and provided with water and normal diet ad libitum. After one week of acclimatization, PICS mice models were established based on the existing literature reports [14]. Forty healthy mice were selected for cecal ligation and puncture (CLP) surgery, while ten mice underwent a sham surgery, designated as the WT group. Within three days after surgery, the acute mortality rate in the surgery group was around 20%, while water and food were provided normally during the period. The surviving mice were randomly divided into three groups: the model group (CK, *n* = 10 mice), the 3 kDa-HA (Zhongshan Biotechnology Co., Ltd, Shandong, China) treatment group (3 K, *n* = 10 mice), and the 1600 kDa-HA (Bloomage Biotechnology Corporation Limited, Hainan, China) treatment group (1600 K, *n* = 10 mice).

On the third day after surgery, the surgical mice survived the acute death period and were used as PICS models for the experiment. We regarded this time as the start of the experiment (0th day), and the mice in the 3 K and 1600 K groups were orally administered with corresponding molecular weight of HA (30 mg/kg) for treatment(Figure 6). The mice in the WT and CK groups were orally administered with the same volume of normal saline. The oral administration treatment started from the third day after surgery, which was the beginning of the experiment, and continued until the end of the experiment.

### 4.2. Monitoring of Mouse Health Status during Treatment

The health status of the mice was monitored during the treatment period. The survival status of each group of mice was recorded daily, and survival curves were plotted from the first day of treatment (3 days post-surgery) to the end of the experiment. Body weight and grip strength of the mice in each group were measured on day 10 and 30 of the experiment.

### 4.3. Collection of Mouse Organs and Histopathological Examination

On the 10th day and the 30th day of the experiment, three mice from each group were randomly selected, anesthetized, and euthanized. The lungs were quickly fixed, and the colon was dissected. The morphology of the intestine was observed, and 1 cm segments of the distal and proximal colon were collected and fixed. The fixed tissues were used to make paraffin sections and stained with hematoxylin and eosin (H&E) to observe histopathological changes. Masson staining was used to observe tissue fibrosis.

### 4.4. Collection and Analysis of Intestinal Microbiota

Fresh mouse fecal samples from each group were collected on day 0, 10, and 30 of the experiment.

Total community genomic DNA extraction was performed using a E.Z.N. A™ MagBind Soil DNA Kit (M5635-02, Omega, New York, NY, USA), following the manufacturer’s instructions. We measured the concentration of the DNA using a Qubit 4.0 (Thermo, Waltham, MA, USA) to ensure that adequate amounts of high-quality genomic DNA had been extracted.

Our target was the V3–V4 hypervariable region of the bacterial 16S rRNA gene. PCR was started immediately after the DNA was extracted. The 16S rRNA V3–V4 amplicon was amplified using 2 × Hieff^®^ Robust PCR Master Mix (Yeasen, 10105ES03, Shanghai, China). Two universal bacterial 16S rRNA gene amplicon PCR primers (PAGE purified) were used: the amplicon PCR forward primer (CCTACGGGNGGCWGCAG) and amplicon PCR reverse primer (GACTACHVGGGTATCTAATCC). The reaction was set up as follows: microbial DNA (10 ng/µL) 2 µL; amplicon PCR forward primer (10 µM) 1 µL; amplicon PCR reverse primer (10 µM) 1µL; 2 × Hieff^®^ Robust PCR Master Mix (Yeasen, 10105ES03, Shanghai, China) (total 30 µL). The plate was sealed and PCR performed in a thermal instrument (Applied Biosystems 9700, Bedford, MA, USA) using the following program: 1 cycle of denaturing at 95 °C for 3 min, first 5 cycles of denaturing at 95 °C for 30 s, annealing at 45 °C for 30 s, and elongation at 72 °C for 30 s; then 20 cycles of denaturing at 95 °C for 30 s, annealing at 55 °C for 30 s, elongation at 72 °C for 30 s; and a final extension at 72 °C for 5 min. The PCR products were checked using electrophoresis in 2% (*w*/*v*) agarose gels in TBE buffer (Tris, boric acid, EDTA) stained with ethidium bromide (EB) and visualized under UV light.

Then, we used the method of constructing the library of 16S gene and then the quantifying and sequencing of the genes. After sequencing, the two short Illumina readings were assembled by PEAR software (version 0.9.8) according to the overlap, and fastq files were processed to generate individual fasta and qual files, which could then be analyzed by standard methods. The effective tags were clustered into operational taxonomic units (OTUs) of ≥97% similarity using Usearch software (version 11.0.667). Chimeric sequences and singleton OTUs (with only one read) were removed, after which the remaining sequences were sorted into each sample based on the OTUs. The tag sequence with the highest abundance was selected as a representative sequence within each cluster. Bacterial OTU representative sequences were classified taxonomically by blasting against the RDP Database and UNITE fungal ITS Database, respectively. Additionally, then we analyzed the statistical and predicated the function of the bacterial.

### 4.5. Statistical Analysis

The α-diversity indices (including Chao1, Simpson, and Shannon indices) were quantified in terms of OTU richness. To assess sample adequacy, rarefaction curves of the observed numbers of OTUs were constructed, and all α diversity indices were calculated with Mothur software (version 3.8.31). The OTU rarefaction curve and rank abundance curves were plotted in R (version 3.6.0). To estimate the diversity of the microbial community of the sample, we calculated the within-sample (alpha) diversity by T test for two groups, and multiple group comparisons were made using ANOVA test. Beta diversity evaluates differences in the microbiome among samples and is normally combined with dimensional reduction methods such as principal coordinate analysis (PCoA), non-metric multidimensional scaling (NMDS), or constrained principal component analysis (PCA) to obtain visual representations. These analyses were visualized using R vegan package (version 2.5–6), and finally, the inter-sample distances were presented as scatterplots. Difference comparison is used to identify features with significantly different abundances between groups using STAMP (version 2.1.3) and LefSe (version 1.1.0). Correlation coefficients and *p*-values between communities/OTUs were calculated using SparCC (version 1.1.0), and correlation matrix heatmaps were drawn using R corrplot package (version 0.84). R ggraph package (version 2.0.0) is used to build network graphs.

Functional prediction analysis of bacteria and archaea was performed using PICRUSt (v1.1.4) software, by comparing existing 16S rRNA gene sequencing data with a microbial reference genome database of known metabolic functions, enabling the prediction of bacterial and archaeal metabolic functions.

Statistical analysis of survival rate, body weight, and grip strength was performed using GraphPad Prism 9.3.1. Data are presented as mean ± SEM (standard error of the mean). Differences between groups were tested for statistical significance by *t*-test. *p* < 0.05 was considered statistically significant.

## 5. Conclusions

This study reveals that HA with different molecular weights can alleviate the progression of PICS in mice to varying degrees. It was found that 1600 kDa-HA can repair the intestinal structure of PICS mice, increase the stability of the gut microbiota structure, and enable them to recover quickly. Although 3 kDa-HA slowly regulates the health status of PICS mice by modulating gut microbiota, the role of this low-molecular size HA in activating inflammatory responses [46] limits its application in PICS therapy. However, previous studies have indicated that HA can be degraded by the gut microbiota [47], and whether low-molecular size HA molecules with triggering marked inflammatory response can be generated during the application of HMW-HA in PICS patients still needs further investigation. In conclusion, our results suggest that treatment with HA mediates its benefit to PICS mice via modulating the gut microbiota, and it may support further clinical investigations on the use of these biopolymers.

## Figures and Tables

**Figure 1 ijms-24-09757-f001:**
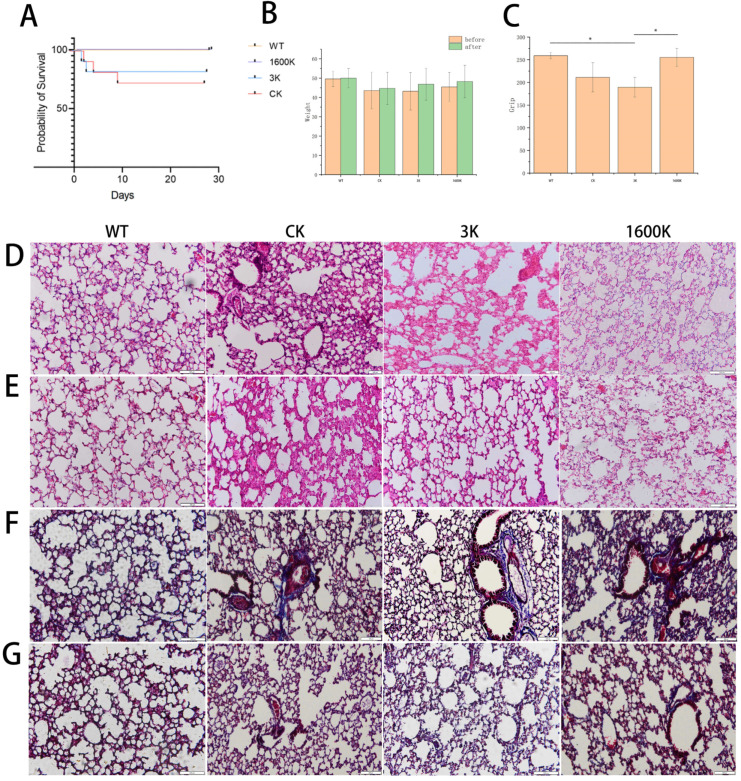
Effect of 3 kDa-HA and 1600 kDa-HA on PICS mice: (**A**) survival rate, (**B**) changes in body weight, (**C**) grip strength on 30th day, (**D**) lung H&E staining results on 10th day, (**E**) lung H&E staining results on 30th day, (**F**) lung Masson staining results on 10th day, and (**G**) lung Masson staining results on 30th day. All representative histological images were taken at a magnification of ×200. Data are presented as mean ± standard deviation, * *p* < 0.05, compared to 3 K and each group. Administration of 1600 kDa-HA improved the survival rate of PICS mice and promoted recovery from lung inflammation. Administration of 3 kDa-HA for 30 days relieved the lung inflammation state of PICS mice.

**Figure 2 ijms-24-09757-f002:**
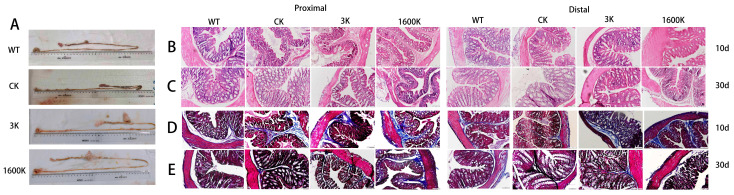
Effect of 3 kDa-HA and 1600 kDa-HA on mice intestinal tissue: (**A**) Intestinal morphology of each group on 10th day, (**B**) intestinal H&E stains results on 10th day, (**C**) intestinal H&E staining results on 30th day, (**D**) intestinal Masson staining results on 10th day, and (**E**) intestinal Masson staining results on 30th day. All representative histological images were taken at a magnification of ×200. The intestinal tissue in the CK group deteriorated severely, while administration of 1600 kDa-HA facilitated rapid recovery of the intestinal structure in PICS mice. Administration of 3 kDa-HA for 30 days relieved the deterioration of the PICS intestine.

**Figure 3 ijms-24-09757-f003:**
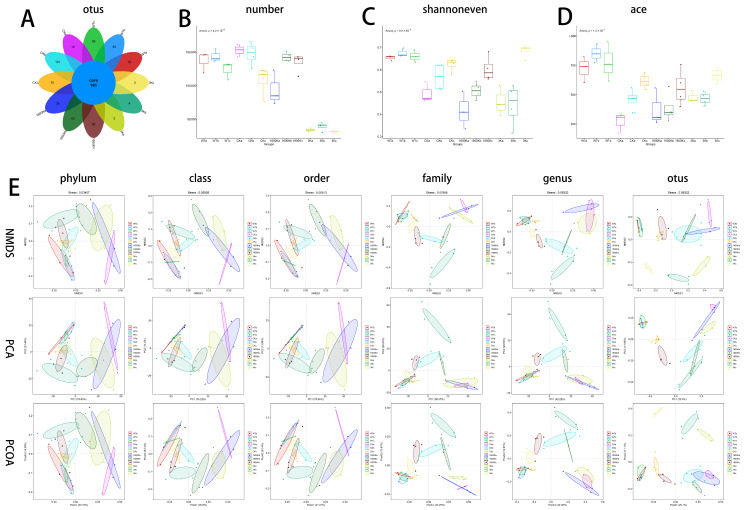
Effects of 3 kDa-HA and 1600 kDa-HA on the abundance and diversity of mice intestinal microbiota. (**A**) Venn diagram of microbial abundance; (**B**) Microbial quantity; (**C**,**D**) Microbial alpha diversity; (**E**) Microbial beta diversity. The letters a, b, and c are used to distinguish between the three different experimental stages: before administration, 10th day of administration, and 30th day of administration. Data are presented as mean ± standard deviation. The microbial quantity in PICS mice gradually recovered except for the 3 K group, but the alpha diversity and beta diversity of the 3 K group were more similar to the WT group.

**Figure 4 ijms-24-09757-f004:**
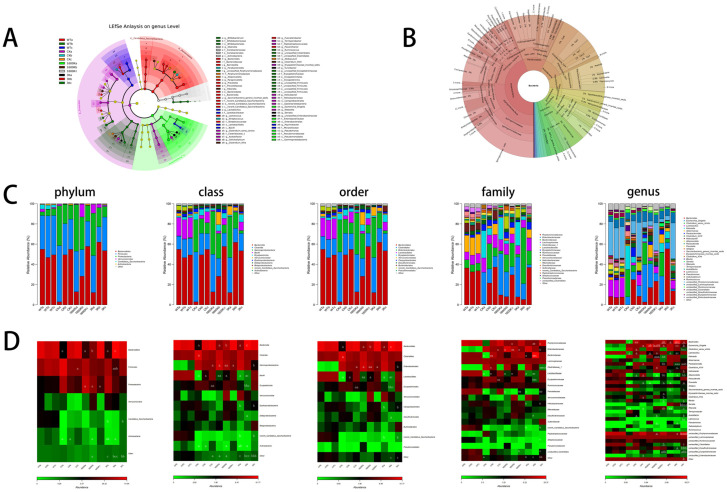
Effects of 3 kDa-HA and 1600 kDa-HA on inter-group differences in mouse intestinal microbiota and the abundance of dominant bacteria at different taxonomic levels were investigated. The study included: (**A**) LEfSe analysis of inter-group differences; (**B**) evolutionary relationships of intestinal microbiota; (**C**) community structure at different taxonomic levels; and (**D**) heatmap results of dominant species at different taxonomic levels. The experimental stages were distinguished by the letters a, b, and c, before administration, on the 10th day, and on the 30th day of treatment, respectively. Data are presented as mean ± standard deviation. ^a^ *p* < 0.05, ^aa^ *p* < 0.01 or ^aaa^ *p* < 0.001 versus WT group; ^b^ *p* < 0.05, ^bb^ *p* < 0.01 or ^bbb^ *p* < 0.001 versus CK group; ^c^ *p* < 0.05 and ^cc^ *p* < 0.01 versus 1600 K group.

**Figure 5 ijms-24-09757-f005:**
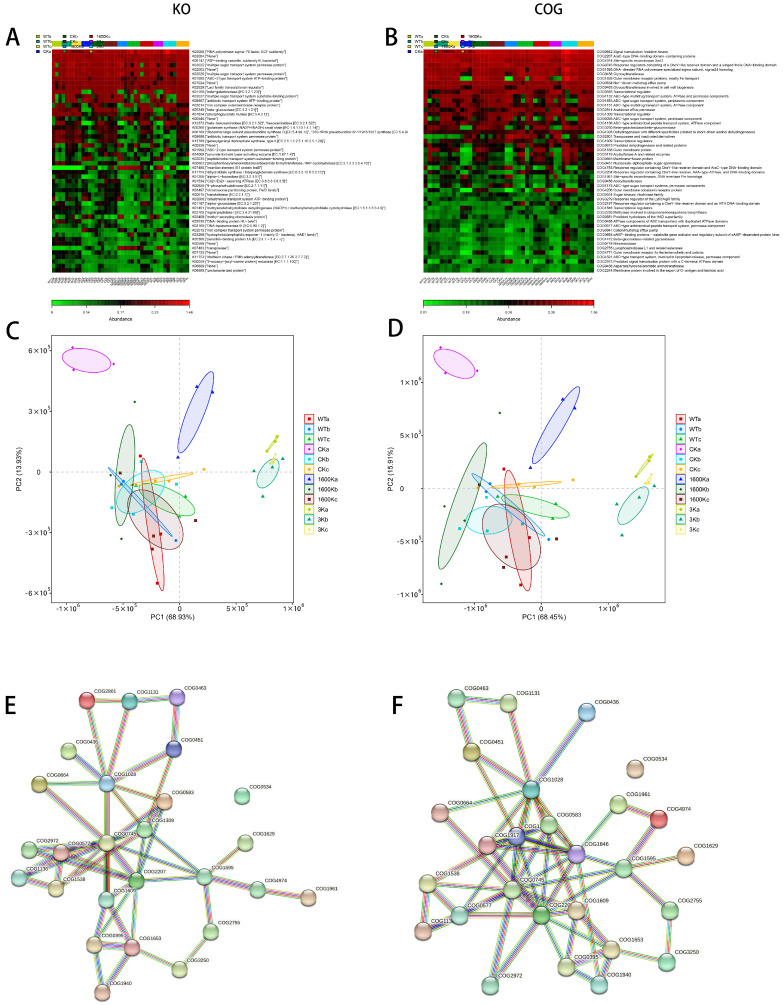
Effects of 3 kDa-HA and 1600 kDa-HA on the function of mouse intestinal microbiota was investigated. The study included: (**A**) heatmap prediction of KO function; (**B**) heatmap prediction of COG function; (**C**) KO function PCA plot; (**D**) COG function PCA plot; (**E**) differential COG pathway analysis comparing 3 kDa-HA with WT and CK groups; and (**F**) differential COG pathway analysis comparing 1600 kDa-HA with WT and CK groups.

**Figure 6 ijms-24-09757-f006:**
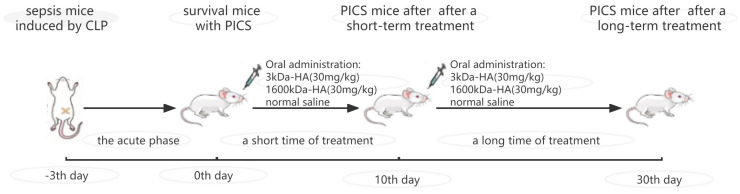
Graphical scheme of timetable for studying the therapeutic effects of 3 kDa-HA (30 mg/kg) and 1600 kDa-HA (30 mg/kg) on PICS mice.

## Data Availability

Not applicable.

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
