# Peer review of "Hyaluronan with Different Molecular Weights Can Affect the Gut Microbiota and Pathogenetic Progression of Post-Intensive Care Syndrome Mice in Different Ways"

_ijms, 2023, doi:10.3390/ijms24119757_

Round 1
Reviewer 1 Report
The paper presented to me for evaluation is an experimental work. It deals with the interesting topic of Post-intensive care syndrome (PICS)
The introduction to the work does not raise my concerns.
The obtained results of the study are presented in a descriptive manner and 5 figures.
Adequate statistical methods were used in the paper.
Attention should be paid to the detailed discussion of the obtained research results.
The paper cites 45 items of current scientific literature.
The conclusions drawn correspond to the obtained research results.
In the classical structure of a scientific paper, the material and methods section should be located after the introduction. I suggest that the order of the various chapters of the publication be changed in accordance with generally accepted standards.
In my opinion, the work is suitable for publication after making the above corrections
Reviewer 2 Report
The manuscript entitled, “HA with Different Molecular Weights can affect the Gut Microbiota and Pathogenetic Progression of PICS Mice in different ways” intends to evaluate the effect of hyaluronic acid with different molecular weights on the gut microbiota of mice model having post-intensive care syndrome. However, some major concerns should be addressed by the authors before any possible consideration of this manuscript is published in the International Journal of Molecular Sciences.
1. The title of the manuscript is vague and the authors should avoid the use of abbreviations in the title of a manuscript.
2. The ‘Abstract/Introduction’ section of the manuscript fails to clarify the aim/objective of this study.
3. What basis is the different molecular weight of HA chosen? Why significant differences in Mw?
4. Intensive Care Syndrome (PICS) is a collection of physical, mental, and emotional symptoms; consideration of just a few physical conditions does not cover the entire range of PISC-associated problems. It would be better if the authors specifically clarified the physiological conditions targeted in this study.
5. As mentioned in the #line_69-70, “…by observing their survival rate and physiological condition as well as the structure and function of gut microbiota at different stages…” What do the authors mean by the term ‘structure and function of gut microbiota’?
6. Except for Figure 1, all the figures are of very small size and poor quality, particularly the legend which should be corrected. According to Fig.1A, 3 KDa decreases survivability, why?
7. Most of the figures are unreadable and could not evaluate the content.
8. The statements written in #line_283-285 are confusing and should be rephrased for clear explanations.
9. The practical application and limitations of this study should be clearly explained in the ‘Conclusion’ section.
Minor editing of English language required
Round 2
Reviewer 2 Report
-